# SpatialPIN: Enhancing Spatial Reasoning Capabilities of Vision-Language Models through Prompting and Interacting 3D Priors

**Chenyang Ma**    **Kai Lu**    **Ta-Ying Cheng**    **Niki Trigoni**    **Andrew Markham**

University of Oxford

chenyang.ma@cs.ox.ac.uk

## Abstract

Current state-of-the-art spatial reasoning-enhanced VLMs are trained to excel at spatial visual question answering (VQA). However, we believe that higher-level 3D-aware tasks, such as articulating dynamic scene changes and motion planning, require a fundamental and explicit 3D understanding beyond current spatial VQA datasets. In this work, we present **SpatialPIN**, a framework designed to enhance the **spatial** reasoning capabilities of VLMs through **p**rompting and **in**teracting with priors from multiple 3D foundation models in a zero-shot, training-free manner. Extensive experiments demonstrate that our spatial reasoning-imbued VLM performs well on various forms of spatial VQA and can extend to help in various downstream robotics tasks such as pick and stack and trajectory planning.

## 1   Introduction

Equipping vision-language models (VLMs) the capacities of spatial reasoning unlocks exciting applications, such as general-purpose reward annotation [52], robotic data generation [61], and grounding 3D object affordances [26, 38]. However, the spatial reasoning capabilities of VLMs on fine-grained spatial understanding tasks are somewhat limited. Current state-of-the-art (SOTA) spatial reasoning-enhanced VLM [12] is mostly tested on spatial visual question answering (VQA), such as determining objects' relative positions and orientations; experiments on higher-level tasks, such as scene comparisons and trajectory planning, which require more nuanced comprehension, are underexplored.

Many works enhance the spatial reasoning capabilities of VLMs by training/fine-tuning them on standard spatial VQA datasets [12]. As a result, VLMs primarily learn surface-level associations between image-text-data triplets. Given the scarcity and difficulty of obtaining spatially rich embodied data or high-quality human annotations for 3D-aware queries, we hypothesize that these VLMs may not generalize to questions outside their dataset distribution or adapt to more challenging tasks that require an advanced level of spatial understanding.

Recent studies [73, 7, 65, 69] in image space understanding show that VLMs, equipped with internet-scale language knowledge, and multimodal foundation models capture complementary knowledge that can be combined to conduct new tasks spanning both modalities without additional training. Given the recent advancements in 3D foundation models [4, 41, 29], this work explores whether there exists an alternative approach to enhance VLMs with higher-level spatial-awareness by incorporating 3D priors from these models.

To this end, we propose **SpatialPIN**, a framework that utilizes progressive prompting and interactions between VLMs and 2D/3D foundation models as "free lunch" to enhance spatial reasoning capabilities in a zero-shot, training-free manner. By using these foundation models to decompose, comprehend,

38th Conference on Neural Information Processing Systems (NeurIPS 2024).

and reconstruct an explicit 3D representation, SpatialPIN grasps the core understanding of the 3D space presented by the 2D image. This allows generalizations to various 3D-aware tasks, from VQAs to 3D trajectory planning.

We provide an extensive empirical study combining multiple off-the-shelf and handcrafted datasets, ranging from fundamental spatial questions regarding relative positions and orientations to providing fine-grained 3D information on objects' locations, sizes, inclinations, and dynamic changes, and plan for robotics tasks with full 3D trajectories. Results show that this straightforward approach significantly outperforms SOTA VLMs trained from extensive spatial VQAs (see SpatialPIN examples in Figure 1), consolidating our belief that a truly 3D-aware VLM can actually be imbued by simply injecting explicit, fundamental knowledge of the 3D scene. With the entire framework being fully modularized, each component can be easily replaced with the latest improvements within its specific domain.

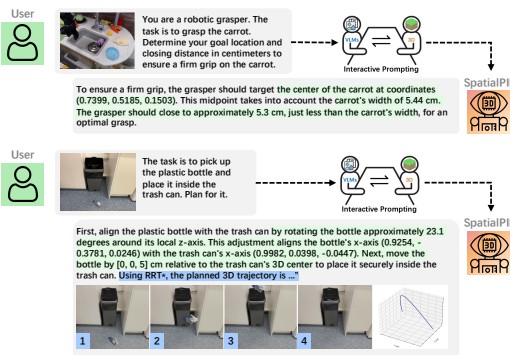

Figure 1: We present **SpatialPIN**, a framework to enhance the **spatial** reasoning capabilities of VLMs through **p**rompting and **in**teracting with 3D priors in a zero-shot, training-free manner.

In summary, our main contributions are threefold:

- We investigate the problem of equipping VLMs with 3D reasoning capabilities without fine-tuning on large spatial VQA datasets.
- We propose SpatialPIN, a modular plug-and-play framework that progressively enhances VLM's 3D reasoning capabilities by prompting and interacting with 3D foundational models.
- We show that SpatialPIN unlocks 3D-aware applications including spatial VQA and both classic and novel robotics tasks, supported by extensive experiments.

## 2 Related Work

**VLM Grounding** With the recent birth of powerful LLMs and VLMs [8, 40, 2], the task of VLM grounding, or combining generative language models with real-world data to adapt to specific cases, has gained significant popularity. Several recent works focused on fine-tuning these LLMs for a wide range of downstream applications, such as interactive decision making [34], multi-task agents [62], or even tasks in interactive environments [67, 11]. A close work to ours is Socratic Model [73], a framework of combining multiple foundation models to unleash LLMs in downstream tasks. However, this work still focuses on tasks in 2D pixel space understanding of images. There remains many challenges in the 3D world to combine information for full scene understanding, which we hope to tackle in our paper.

**VLM Spatial Understanding** Many VLMs encompass the ability of image-space reasoning and understanding [13, 33, 40]. There are even efforts in incorporating these understandings into image space manipulations and editing [7, 65]. However, the current ability of VLMs to fully understand a 3D scene and the potential interactions within this scene is still rather limited. Several works build from this foundation and establish datasets to help with spatial reasoning/understanding [30, 39, 45]. Recently, SpatialVLM [12] proposed fine-tuning a VLM on 3D-VQA datasets to enhance the precision of VLMs on 3D understanding tasks. Nevertheless, using a 3D-VQA dataset only provides a partial picture to the complete 3D understanding of an image, and could lead to suboptimal performances under out-of-distribution tasks. In this work, we hope to introduce holistic 3D information from multiple 3D foundation models via prompting and interactions as a way to enhance VLMs with a comprehensive 3D understanding given RGB inputs.

## 3 Method

Given an RGB image $I \in \mathbb{R}^{H \times W \times 3}$ of a scene with $K$ unknown objects and a spatial task $Q$, our goal is to inspire VLMs with spatial reasoning capabilities and solve $Q$ with fine-grained 3D understanding.

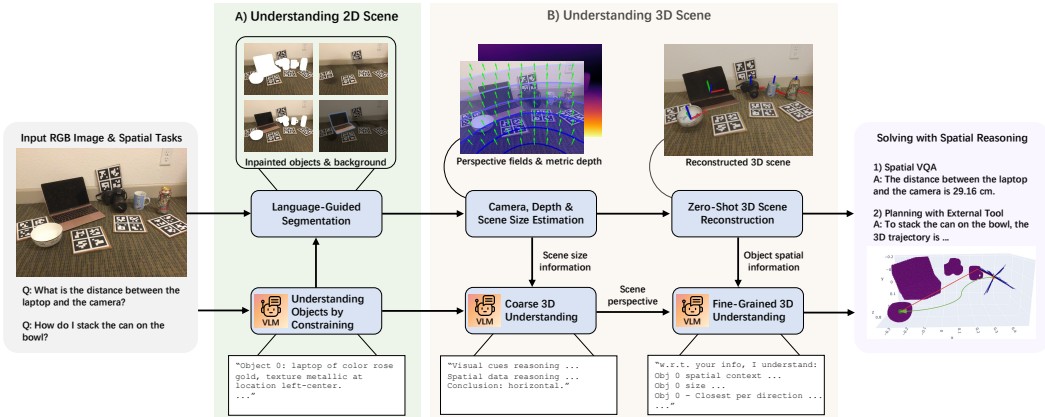

Figure 2: **SpatialPIN**. Our plug-and-play framework is fully modularized and designed for zero-shot deployment. Each module can be easily replaced with the latest updates. Exact prompts for VLMs are in Appendix.

To prevent the models from overfitting to the standard problems from spatial VQA datasets [12], we hope to derive a method that utilizes fundamental 3D foundation models to provide explicit scene understandings, then leverage the generalization capabilities of VLMs to tackle unforeseen tasks—all within a zero-shot, training-free manner.

Our modular pipeline, SpatialPIN, enhances VLMs' spatial understanding of an image through progressive interactions with the scene decomposition, comprehension, and reconstruction processes with prompting. For image scene understanding (Sec. 3.1), we use VLM to describe objects by appearance and 2D location, complemented by language-guided segmentation and repainting models to obtain occlusion-free object masks. Elevating 2D understanding to coarse 3D (Sec. 3.2), we use metric depth estimation and perspective fields to estimate the 3D scene size and conduct perspective canonicalization with VLM. For fine-grained 3D understanding (Sec. 3.3), we partially reconstruct the 3D scene, with the full 3D representation of foreground objects and the background as a plane. With the reconstructed 3D scene, we summarize spatial information and prompt it to the VLM for various downstream tasks.

### 3.1 2D Image Scene Understanding

**Prompting: Objects Understanding by Constraining**   We start with querying VLM to identify and understand objects given $I$. We explicitly ask VLM to describe the objects by precise color, texture, and 2D spatial locations. This step is vital for two reasons: 1) enhance VLM's understanding of the objects, 2) differentiate between items of similar or identical categories and appearances.

As a concrete example, given the left image of Fig. 2, VLM outputs: "

```
object 0:  laptop of color rose gold, texture metallic at location left-center.
object 1:  camera of color black, texture smooth at location center-right.  ..."
```

**2D Representations Refinement**   The concise descriptions of identified objects are used as input text prompts for a language-guided segmentation model, enabling the acquisition of $K$ segmentation masks $\{M_k^{occ}\}_{k=1}^K$, with each mask corresponding to a unique object.

However, an object $i \in [1, K]$ may be occluded by other object(s), leading to an incomplete mask $M_i^{occ}$, which may be burdensome when we elevate the image to a 3D representation in the later stage. To resolve this, we create an inpainting mask, $M_i^{inp}$, for each object, in which all objects except the one itself are removed and replaced with white pixels. The inpainted masks are again fed to the language-guided segmentation model along with input text prompts such that occlusion-free object masks, $\{M_k^{of}\}_{k=1}^K$, are obtained. This two-step segmentation process for object $i$ is formulated as:

$$M_i^{occ} = \text{seg}(I, \tau_i), \quad M_i^{of} = \text{seg}(\text{inpaint}(M_i^{inp}), \tau_i), \tag{1}$$

where seg denotes the language-guided segmentation and $\tau_i$ denotes the description of object $i$. In practice, to cleanly remove objects without residual fragments for inpainting, we apply dilation to and expand the white areas. Inpainted background $I_{bg}$ is acquired by removing and replacing all objects with dilated white pixels.

## 3.2 Coarse 3D Scene Understanding

**Scene Size Estimation**   Using the estimated metric depth [29] and estimated camera intrinsic matrix by finding field of view (FOV) through perspective fields [29], we backproject to determine the dimensions of the 3D spatial scene.

**Prompting: Perspective Canonicalization**   3D information without any knowledge regarding the camera perspectives lead to ambiguities [12]. Consider a question "What is the orientation of the bowl relative to the laptop?" with the input scene in Fig. 2, but taken from a top-down perspective. VLMs may output ``downward to the left'', but the correct answer should be "front-left" because humans perceive orientation from a horizontal angle. To address this, we provide the VLM with $I$, estimated scene size, and maximum and minimum dimensions, allowing it to reason about the camera shot angle (horizontal/top-down/bottom-up). Scene size information helps differentiate shot angles by providing clues about spatial layout and object proportions. For instance, if the depth variation is small, the VLM can infer a top-down or bottom-up angle along with visual cues.

As a concrete example, given the left image of Fig. 2, VLM outputs:

``Visual cues reasoning:  Objects are viewed from the side, indicating the camera is positioned horizontally with a slight elevation.

Spatial data reasoning:  The depth varies significantly from 57.50 cm to 115.00 cm, indicating the camera captures the scene across different distances, supporting a horizontal perspective.

Conclusion:  horizontal.''

## 3.3 Fine-Grained 3D Scene Understanding

We partially reconstruct the 3D scene with full representation of foreground objects while simplifying the inpainted background as a plane, as shown in Fig. 3(a). We summarize spatial information from the reconstructed scene and prompt it to the VLM. Please see our Appendix for implementation details about reconstruction.

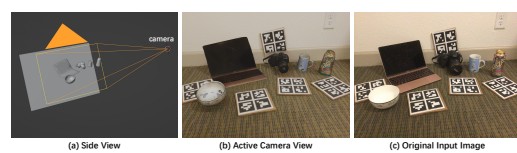

(a) Side View    (b) Active Camera View    (c) Original Input Image

Figure 3: Our method of **partial 3D scene reconstruction (a).** The reconstructed scene (b) and the input image (c) show high alignment.

**Scene Initialization**   Given the occlusion-free object masks, $\{M_k^{of}\}_{k=1}^K$, we use single-view 3D reconstruction model [41] to acquire object 3D models, $\{O_k\}_{k=1}^K$, with canonical poses determined during reconstruction. Pinhole camera is set at the origin, looking at positive depth-axis. With the estimated background plane size (Sec. 3.2), we move the background plane, $O_{bg}$ (visually identical to $I_{bg}$), along the depth-axis to fit precisely within the camera.

**Scene Reconstruction**   To resolve the imprecision of backprojection, our goal is to position object 3D models into the reconstructed 3D scene without visual discrepancies and ensure accurate depth. Instead of using naive backprojection, for an object $i \in [1, K]$, we perform raycasting from object 3D center $t_i^c$ on the camera plane to object 3D center $t_i^{bg}$ on the background plane with metric depth $d_i$. The 3D coordinate $t_i$ of object $i$ is:

$$d_i = \left| I_{dep}(\text{center}(M_i^{of})) \right|, \quad t_i = t_c + \frac{d_i}{\left| t_i^{bg} - t_i^c \right|} \times (t_i^{bg} - t_c). \tag{2}$$

The rotation $R_i$ of the 6D pose of object $i$, $P_i = [R_i \mid t_i]$ is explained previously. After integrating all 3D object models into the 3D scene, we refine each object's scale to accurately reflect depth variations by rendering binary masks and evaluate the length of their contour lines relative to their occlusion-free masks, through the lens of the pinhole camera, $t_c$.

We determine the principal axes (x-axis, y-axis, and z-axis) of each object using the minimal oriented bounding box (OBB), which is essential for unlock novel applications.

**Prompting: Objects and Spatial Context Understanding**   The reconstructed 3D scene from $I$ with accurate object poses and scales is denoted as $V_0$. As the final step of progressive prompting, we feed VLM the fine-grained 3D information derived from $V_0$, grounding on the canonicalized perspective (Sec. 3.2). For example, with the input image in Fig. 2 and a horizontal camera shot angle, depth corresponds to the positive y-axis (similarly, in a top-down/bottom-up view, depth is

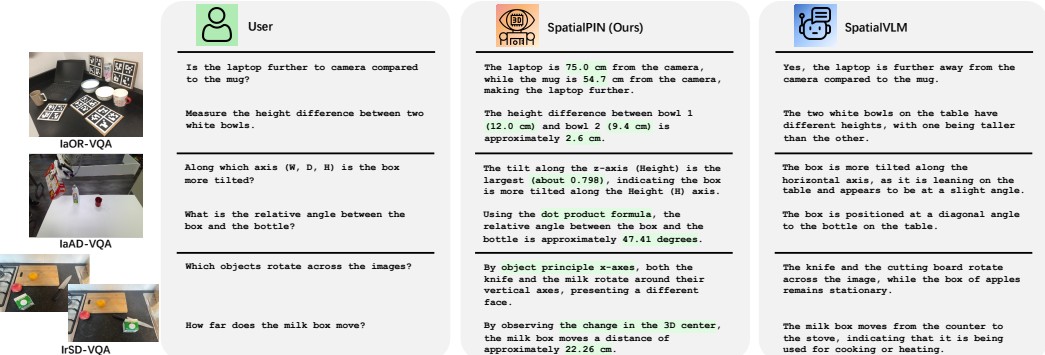

Figure 4: **Qualitative examples of spatial VQA.** SpatialPIN outputs answers with fine-grained 3D reasoning. *Zoom in for better view.*

the negative/positive z-axis) in a right-handed coordinate system. The width and height axes can be determined accordingly, aligning each axis's orientation with human perception.

We feed VLM a paragraph describing the objects' poses, sizes, and principal axes in physical units, alongside their spatial relationships. To augment VLM's understanding, we also feed $V_0$ with visualized object axes (see Fig. 2B). Visualizing 3D spatial information is pivotal in improving VLMs' understanding of 3D spatial contexts derived from 2D images, validated by 3DAxiesPrompts [37]. Yet, we want to **emphasize** that we do not feed hardcoded information, such as objects' relative distances and inclinations, to VLMs. Instead, we aim for the summarized 3D information to enhance VLMs' general spatial understanding.

As a concrete example for the left image on Fig. 2:

```
``Obj 1 spatial context:  3D center:  [7.0, 100.0, 9.0] cm; X-axis (right):  [0.9529, -0.2456, 0.1779];
Y-axis (back):  [-0.3528, 0.8746, 0.3327]; Z-axis (up):  [-0.1761, -0.3285, 0.9279]
Obj 1 size:  13.54 cm x 9.37 cm x 9.50 cm (WxDxH)
Obj 1 closest per direction:  left:  Obj 0; right:  Obj 2 ...''
```

### 3.4 Combining External Tools for Downstream Tasks

By partially reconstructing the 3D scene with visual alignments, our framework enables VLMs to use tools like rapidly-exploring random tree star (RRT*) [31] to generate accurate, collision-free paths based on task specifications (more details in Appendix). This capability unlocks novel and interesting applications when combined with task-specific prompting techniques, shown in Experiments (Sec.4).

## 4 Experiments

We conduct experiments to answer the following questions: 1) Does our framework enhance the general spatial reasoning capabilities of VLMs, and how well does it perform? 2) What novel applications does our framework unlock for VLMs, and how well do we perform in these applications? 3) How effective is each module in our framework?

Since we evaluate our approach on a wide range of tasks to test VLMs' higher-level spatial awareness, some tasks are novel and lack existing/open source datasets. Therefore, for all our experiments, we use a combination of 4 existing datasets and 2 hand-crafted datasets.

**Implementations** The language-guided segmentation model is Language Segment-Anything [44] and the repainting model is LaMa [55]. We use One-2-3-45++ [41] for single-view 3D reconstruction, perspective fields [29] for camera intrinsic estimation, and ZoeDepth [4] for depth estimation. For partial 3D scene reconstruction, we use Blender [17] as the 3D software. All inference is run on 1 NVIDIA A10 GPU with 24GB RAM.

### 4.1 Spatial Visual Question Answering

We experiment on the basic form of spatial VQA introduced by SpatialVLM (IaOR-VQA), and two new forms introduced by us (IaAD- & IrSD-VQA). For IaOR-VQA, please check SpatialVLM [12] for details. For IaAD- & IrSD-VQA, please see our Appendix.

**Intra-Image Object Relations VQA (IaOR-VQA)**  As the basic form of spatial VQA, it involves spatial reasoning about object relative orientations and sizes. This is divided into qualitative (e.g., "is [A] in front of [B]", "is [A] smaller than [B]") and quantitative (e.g., "how far apart are [A] and [B]", "measure the width of [A]") questions.

We follow the evaluation method of SpatialVLM [12]. Since SpatialVLM did not release their evaluation dataset, we reproduce one using RGBD images from NOCS [57] (object dataset), RT-1 [6], and BridgeData V2 [56] (robotics manipulation datasets). We sample 13, 20, and 20 distinct scenes from each. We generate QA pairs using the SpatialVLM data generation pipeline [51], followed by manual refinement. We check correctness for qualitative questions and calculate distances for quantitative questions. We annotate 300 qualitative and 200 quantitative spatial VQA pairs (SpatialVLM has 331 and 215 for each).

**Intra-Image Angular Discrepancies VQA (IaAD-VQA)**  We propose a new form of Spatial VQA that needs spatial reasoning about objects' inclinations. It includes qualitative (e.g., "is [A] tilted", "is [A] more tilted than [B]") and quantitative questions (e.g., "how many degrees is [A] tilted vertically", "measure the angle between [A] and [B]").

Since this form of Spatial VQA involves out-of-plane rotations, YCBInEOAT [64] (object tracking dataset) is a suitable choice. We sample 30 scenes from it and annotate 50 questions each for qualitative and quantitative spatial VQA pairs.

**Inter-Image Spatial Dynamics VQA (IrSD-VQA)**  We further propose a more challenging form of Spatial VQA. Given two images with multiple objects, the objects in the second image may move, rotate, incline, or the image may have a change in camera angle. The VLM needs to reason about these changes. Example qualitative questions include "does [A] move, rotate, or incline", "does [A] incline along the y-axis" while quantitative questions include "how far does [A] move", "how many degrees does [A] rotate horizontally".

As it is difficult to find a dataset that meets these requirements, we craft our own. We capture 20 image pairs using an iPhone 12 Pro Max, with each image containing $1-5$ objects, and annotate 50 questions each for qualitative and quantitative spatial VQA pairs.

**Results**  The results in Tables 1 and 2 on qualitative and quantitative IaOR-VQA demonstrate that providing various VLMs fine-grained 3D information enhances their spatial reasoning capacities by a large margin. Surprisingly, VLMs with math and geometry reasoning capacities (e.g., GPT-4V, GPT-4o) show substantial improvements with this information.

Table 1: **Qualitative IaOR-VQA.** We exclude comparisons to PaLI [14], PaLM-E [20], and PaLM 2-E [3] as they are not open source, and include experiments with GPT-4o [1] in addition to GPT-4V [47], LLaVA-1.5 [40], and InstructBLIP [18]. We use the HF version of SpatialVLM [51].

|            | GPT-4V | | GPT-4o | | LLaVA-1.5 | | InstructBLIP | | SpatialVLM |
|------------|---------|--------|---------|--------|-----------|--------|--------------|--------|------------|
|            | w/o ours | w ours | w/o ours | w ours | w/o ours | w ours | w/o ours | w ours | |
| Accuracy % | 70.7 | 86.3 | 69.0 | **87.3** | 70.0 | 83.0 | 62.3 | 79.3 | 76.7 |

Table 2: **Quantitative IaOR-VQA.** SpatialVLM measures the accuracy by the percentage of answers that fall within 0.5x to 2.0x of the ground truth value. We also evaluate within narrower ranges of 0.75x to 1.33x and 0.9x to 1.11x. "Output number" means VLMs produce number in the response instead of vague descriptions.

|                       | GPT-4V | | GPT-4o | | LLaVA-1.5 | | InstructBLIP | | SpatialVLM |
|-----------------------|---------|--------|---------|--------|-----------|--------|--------------|--------|------------|
|                       | w/o ours | w ours | w/o ours | w ours | w/o ours | w ours | w/o ours | w ours | |
| Output numbers %      | 0.8 | 98.5 | 31.5 | **99.5** | 23.5 | 97.0 | 28.5 | 98.5 | 91.0 |
| In range [50, 200] %  | 0.0 | 73.5 | 14.0 | **74.5** | 16.5 | 43.0 | 8.0 | 31.5 | 33.5 |
| In range [75, 133] %  | 0.0 | 69.5 | 8.5 | **70.5** | 6.0 | 29.0 | 2.5 | 22.0 | 20.5 |
| In range [90, 111] %  | 0.0 | 54.5 | 3.0 | **55.0** | 2.0 | 14.5 | 0.0 | 11.5 | 7.5 |

The results in Tables 3 and 4 demonstrate the effectiveness of our approach on both qualitative and quantitative IaOR-VQA and IrSD-VQA tasks. Notably, the performance on quantitative IaOR-VQA is suboptimal compared to quantitative IrSD-VQA, despite the latter being more challenging. We

Table 3: **Qualitative IaAD-VQA** & **IrSD-VQA.** Since we test SpatialPIN on one VLM backbone for our proposed spatial VQA, for fair comparison, we should use SpatialVLM backbone (PaLM 2-E [3]). However, since it is not open source, we use GPT-4o as our backbone, as it shows the most improvement with our framework.

| | IaAD-VQA | | | IrSD-VQA | | |
|---|---|---|---|---|---|---|
| | GPT-4o | GPT-4o + ours | SpatialVLM | GPT-4o | GPT-4o + ours | SpatialVLM |
| Accuracy % | 68 | **84** | 62 | 64 | **82** | 54 |

Table 4: **Quantitative IaAD-VQA** & **IrSD-VQA.**

| | IaAD-VQA | | | IrSD-VQA | | |
|---|---|---|---|---|---|---|
| | GPT-4o | GPT-4o + ours | SpatialVLM | GPT-4o | GPT-4o + ours | SpatialVLM |
| Output numbers % | 38 | **100** | 66 | 30 | **100** | 78 |
| In range [50, 200] % | 8 | **64** | 12 | 10 | **68** | 26 |
| In range [75, 133] % | 2 | **42** | 6 | 4 | **54** | 12 |
| In range [90, 111] % | 0 | **30** | 2 | 2 | **38** | 4 |

believe this is because, for quantitative IrSD-VQA, the VLM sometimes confuses the camera and world coordinate frames, comparing the object's principal axes with the world axes to reason about changes in angles.

Fig. 4 presents qualitative examples on all forms of spatial VLM.

## 4.2 Robotics Pick and Stack

Pick and stack is a classic robotics task. Given a robot's egocentric observation of a scene with multiple objects and a task description, our pipeline uses traditional planning to solve the problem. This task demands advanced spatial reasoning, as the model must comprehend 3D locations, sizes, and physical properties of the objects (i.e., how much to grasp and how high to drop? Is the object deformable or articulated so the robotic grasper needs to grasp more firmly?). For instance, grasping and stacking a soft toy bear on a cube is significantly different from stacking a solid apple on a mug. The model reasons about grasping and stacking policies, directly outputting 3D trajectories for the robot's end effector using traditional path planning algorithm as external tool.

**Set-Up** We set up the pick-and-stack problem in the ManiSkill [22] simulator, applying real-world physics properties. Rigid and articulated objects are chosen from the YCB dataset [10] and are randomly allocated on the table within the robotic arm's reach, with observations from different perspectives. We create 50 scenes. Since robot observations from simulated scenes suffer from sim2real gap and consider that most real-world robots have depth sensors, we use ground truth camera matrix and depth.

We compare our method to the following baselines: 1) direct 3D information output from our framework without GPT-4o [1] reasoning about physics and object properties and 2) SpatialVLM with our RRT* trajectory generation module.

**Results** Table 5 shows the results, with a qualitative example demonstrated in Fig. 5. The results indicate that using precise 3D information from our framework significantly improves the success rate, and incorporating VLM reasoning further enhances performance.

## 4.3 Discovering and Planning for Robotics Tasks from a Single Image

We present a novel task that requires advanced spatial reasoning capacities of VLMs. Given a single RGB image of any scene comprising unknown environments and objects, the VLM discovers potential tasks and plans their execution with full 3D trajectories, with the **motivation** that it can be used for robot learning in future research. To solve this complex task and visualize the execution using our framework, we introduce: 1) a task proposal approach using VLM, 2) a novel axes-constrained 3D planning approach that enables spatial reasoning-imbued VLM to plan the object motion based on the proposed tasks by specifying waypoints. Please see Appendix for the pipeline and details.

**Dataset** We create a diverse evaluation dataset by combining self-captured photos (38) using an iPhone 12 Pro Max and scenes (13) from NOCS [57]. Our dataset covers diverse scenes (*e.g.*, office,

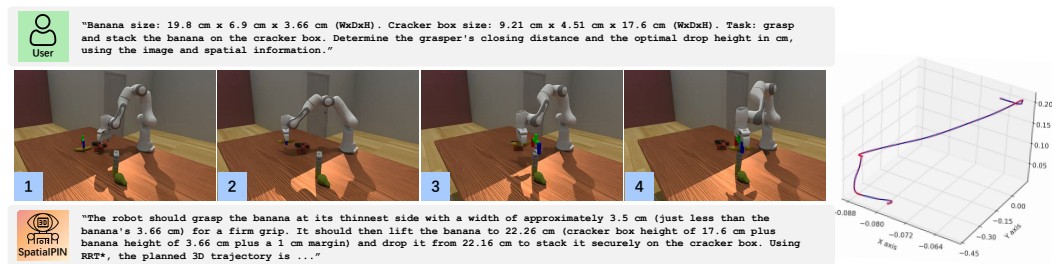

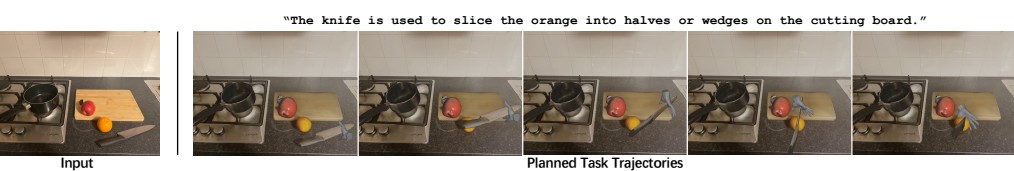

Figure 5: **Qualitative examples of pick and stack (top) and task trajectory planning (bottom).** SpatialPIN successfully outputs picking and stacking policies using spatial reasoning and plans 3D trajectories with geometric awareness to align with task descriptions.

Table 5: **Pick and stack.** We classify the success rates into: 1) successfully picked, 2) successfully picked and contacted the target object but slipped/collided, and 3) successfully picked and stacked.

|  | GPT-4o + ours | Our Direct 3D Info | SpatialVLM + RRT* |
|---|---|---|---|
| Picked % | **44** | 28 | 16 |
| Picked & contacted % | 8 | 12 | 6 |
| Picked & stacked % | **36** | 16 | 10 |

kitchen, bathroom), and features a rich diversity of object categories (116) and quantities (185), with each image containing $1 - 7$ objects and $1 - 3$ tasks proposed for each object (278 tasks/planned trajectories in total). The dataset's diversity is further enhanced by the variety of perspectives (*e.g.*, frontal, top-down, side views). This deliberate choice of diverse angles, both in our own image capturing process and through the random extraction of frames from NOCS, aims to simulate a realistic and challenging array of scenes for evaluation. See Appendix for statistics and visuals.

**Qualitative Demonstration**   We present a qualitative example in Fig. 5. Additional examples in Appendix shows our framework's capability to produce diverse and accurate task trajectories spanning various scenes and tasks.

**Human Evaluation: User Study**   We rely on human preference evaluation as one of our quantitative metrics. We ask 25 users to rate 5 translation and 5 rotation task executions in terms of task description alignment. For these complex context-dependent manipulation tasks, we instead ask users to judge 10 executions relative to human action, and to encapsulate their perception of the action in our with a single sentence. These sentence description will be used to test human understanding of our planned trajectories (please see Appendix). Note that our user study size is similar to those representative works such as ControlNet [74] and Prompt-to-Prompt [24]. Results in Table. 6.

Table 6: **User study.** Ratings (scale $1 - 5$) are averaged.

|  | Rating ↑ |
|---|---|
| Rotation | 4.58 |
| Translation | 4.43 |
| Manipulation | 4.29 |

**Machine Understanding**   We assess the interpretability of our generated task executions from a machine's perspective using SOTA video understanding model, Video-LLaVA-7B [35]. We use two approaches: binary classification and descriptive generation. For classification, we feed the model with the task descriptions generated by VLM and ask question (is the video doing...?). In generation, we prompt Video-LLaVA-7B to articulate its interpretation of our task executions. To quantify the correspondence between the model's perception and the tasks, we use OpenCLIP cosine similarity score [15].

Table 8: **Ablation study.** For quantitative IaOR-VQA, the accuracy is measured by the answers that fall within 0.75x to 1.33x of the ground truth value.

| | Overall Design | | | 2D Understanding | 3D Understanding | | | Ours |
|---|---|---|---|---|---|---|---|---|
| | ShAPO | SAM-6D + 3D models | SpatialVLM | w/o objects | w/o coarse | w/o fine-grained | w/o both | |
| Qualitative | 36.7 | 48.0 | 81.3 | 68.3 | 76.0 | 63.3 | 61.7 | **87.3** |
| Quantitative | 29.5 | 37.0 | 62.5 | 54.5 | 64.5 | 50.5 | 58.0 | **70.5** |

However, we find that even SOTA video understanding model shows limited performance. To assess false positive rate in classification, we deliberately misalign the sequence of generated task executions with their corresponding task descriptions, expecting a theoretical accuracy of 0%. Contrary to expectations, Video-LLaVA-7B reports a false positive rate of 36.3%. To adjust for this anomaly, we subtract this rate from the model's raw accuracy for correctly aligned video-task pairs. This method, while unconventional, provides a more fair and reasonable evaluation of machine video understanding, underscoring the current challenges faced by video understanding models in accurately interpreting complex video content. Results in Table. 7.

Table 7: Results for machine understanding (classification and generation) on 278 task executions.

| | Machine |
|---|---|
| Raw Acc ↑ | 0.974 |
| Fal-Pos Rate ↓ | 0.363 |
| True Acc ↑ | 0.611 |
| OpenCLIP ↑ | 0.636 |

## 4.4 Ablation Study

We evaluate the effectiveness of each module in our framework on IaOR-VQA by 1) seeking alternative designs of the overall pipeline and 2) removing each component in our ablations.

**Overall Design** To demonstrate our framework's generalization across a wide range of objects, We replace our 2D + 3D pipeline with: 1) SOTA mesh-free single image object pose and size estimation model, ShAPO [27], 2) SOTA mesh-based single image object pose and size estimation model, SAM-6D [36], and feeds it with the object 3D model reconstructed by One-2-3-45++ [41], and 3) the data generation backbone of SpatialVLM [12]. Since models 1) and 2) do not provide language annotations for their outputs, we first summarize the numerical outputs using our approach in Sec. 3.3. Then, GPT-4V identifies QA pairs.

**Removing 2D Understanding Module** In this case, the VLM no longer examines the objects through prompting, and only the object name is input into the language-guided segmentation model.

**Removing 3D Understanding Modules** This means there is no scene size estimation, and the VLM does not conduct perspective canonicalization. During 3D scene reconstruction, we assume the image plane width to be 1 meter.

To validate the fine-grained 3D scene understanding module, we replace object mask raycasting with backprojection using the object's 2D center and remove the object scale calibration.

To demonstrate the overall effectiveness of our 3D understanding modules, we simply backproject the input image with the estimated metric depth.

**Results** Table 8 demonstrates the effectiveness of each module in our framework. The results also highlight the limitations of using off-the-shelf SOTA mesh-free and mesh-based single-image object pose and size estimation methods as our backbone. These methods are not language-driven and may struggle to generalize to novel objects in diverse input scenes.

## 5 Discussion and Conclusion

We present **SpatialPIN**, a framework designed to enhance the **spatial** reasoning capabilities of VLMs through **p**rompting and **in**teracting with 3D priors in a zero-shot, training-free manner. We see our work as a step towards equipping VLMs with more generalized spatial reasoning capacities, demonstrated through applications in various forms of spatial VQA and both traditional and novel robotics tasks.

**Limitations** Readers may be curious about the inference speed of our framework. The bottleneck is the 3D object reconstruction process and the API call to closed-source VLMs ($\sim 20$ seconds per image). However, we want to highlight that this process runs only once per image, and the speed is expected to improve with future versions of 3D foundation models.

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

# Appendix for *SpatialPIN*

## A  Overview

This Appendix includes: 1) more technical details about our partial 3D scene reconstruction, 2) additional details, templates, and visualizations of our proposed two forms of SpatialVQA, 3) implementation details on our proposed application: discovering and planning for robotics tasks from a single image (task proposal, axes-constrained motion planning through waypoints, and trajectory generation and smoothing), 4) more experiments on our proposed application (dataset statistics, additional qualitative demonstrations, human understanding, and task diversity), and 5) prompt details for VLMs.

## B  Partial 3D Scene Reconstruction Details

**Metric Depth Estimation**  Because a significant portion of depth estimation model [70, 50, 4, 5, 68] is trained on depth datasets with depth data determined by sensors [21, 53] and stereo matching [71, 16, 58, 59, 66], we assume that the predicted normalized depth is the perpendicular distance to the camera plane, instead of a straight line from the object to the camera lens [32].

**Camera Intrinsic Estimation**  Given an RGB image, With the estimated vertical field of view (FOV), $\theta_v$, through perspective fields [29], the camera focal length $f$ can be found by:

$$f = \frac{H}{2 \tan\left(\frac{\theta_v}{2}\right)}, \tag{3}$$

where $H$ is the image height in pixels.

**Object 6D Pose Estimation**  Single-view 3D reconstruction model reconstructs mesh $O_i$ of object $i$ at the 3D origin, in the coordinate frame set by the input mask $M_i^{of}$, and captures $O_i$ by a pinhole camera. This camera, with 6D pose $P_c = [R_c \mid t_c^{origin}]$, captures $O_i$'s canonical pose within the image. Thus, we can restore all objects' canonical poses across all images by identifying $P_c$.

We use One-2-3-45++ [41], which provides the camera pose. For models without this information, we develop an efficient method to determine the pose by comparing object masks with rendered 3D model templates, inspired from matching-based 6D pose estimation works [36, 46, 9, 42].

We generate a set of object templates, denoted as $\{T_j^i\}_{j=1}^J$, each rendered from the object's 3D model $O_i$. These templates are created by positioning the camera at various locations on an icosphere surrounding the object in $SE(3)$ space, which simulates a spherical coverage around the object to capture its geometry from all angles uniformly. For each template $T_j^i$, we compute a matching score against the occlusion-free object mask $M_i^{of}$.

We propose a simple yet effective score matching method. We draw a bounding rectangle around the segmented object inside $M_i^{of}$ and across all $\{T_j^i\}_{j=1}^J$, and crop the bounding rectangle. We then calculate the shape similarity between the contour line of cropped $M_i^{of}$ and that of each cropped $T_j^i$ using Hu moments [25]. Additionally, we crop and resize the bounding rectangle to the same dimension, and evaluate the similarity based on the pixel area of the cropped and resized masks. Our score matching method can be formulated as:

$$m_i^{A,h} = \text{Hu}(\text{findContour}(\text{crop}(M_i^{of}))), \qquad pa_i^A = \text{sum}(\text{resize}(\text{crop}(M_i^{of}))),$$
$$m_{i,j}^{B,h} = \text{Hu}(\text{findContour}(\text{crop}(T_j^i))), \qquad pa_{i,j}^B = \text{sum}(\text{resize}(\text{crop}(T_j^i))),$$
$$\mathcal{L}(M_i^{of}, T_j^i) = \alpha \left| 1 - \frac{\min(pa_i^A, pa_{i,j}^B)}{\max(pa_i^A, pa_{i,j}^B)} \right| + \beta \sum_{h=1}^{7} \left| \frac{1}{\text{sgn}(m_i^{A,h}) \cdot \log(m_i^{A,h})} - \frac{1}{\text{sgn}(m_{i,j}^{B,h}) \cdot \log(m_{i,j}^{B,h})} \right|. \tag{4}$$

This dual approach allows for a comprehensive comparison that incorporates both the geometric configuration and the scale of the object representations. The best-matched template can be found by $\arg\min_{j=1}^J \mathcal{L}(M_i^{of}, T_j^i)$.

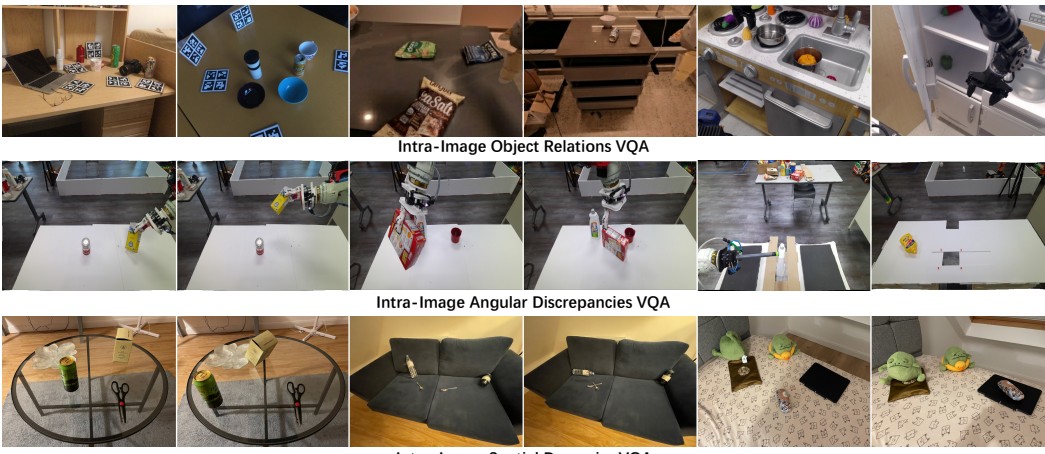

Intra-Image Object Relations VQA

Intra-Image Angular Discrepancies VQA

Inter-Image Spatial Dynamics VQA

Figure 6: Example input images of all forms of spatial VQA.

**Object Scale Calibration** After integrating all 3D object models $\{O_k\}_{k=1}^{K}$ into the 3D scene, each with a pose $\{P_k = [R_k \mid t_k]\}_{k=1}^{K}$ and an initial scale $\{S_k^{init}\}_{k=1}^{K}$ set by the single-view 3D reconstruction model, we refine their scales to accurately reflect depth variations (*e.g.*, moving an apple from close to the camera to a distant corner reduces its apparent size). Through the lens of the pinhole camera with pose $P_c$, we render binary masks for each object and evaluate the length of their contour lines relative to their occlusion-free masks. The adjusted, final scale of object $i$ can be expressed as:

$$S_i^{adj} = S_i^{init} \times \frac{\text{arcLength}(\text{findContour}(M_i^{of}))}{\text{arcLength}(\text{findContour}(M_i^{rend}))}, \tag{5}$$

where $M_i^{rend}$ is the rendered mask of object $i$.

## C    Additional Experiments and Details

### C.1    Spatial Visual Question Answering

**Intra-Image Angular Discrepancies VQA (IaAD-VQA)** For annotation, since YCBInEOAT [64] offers ground truth object 6D poses, we first determine the table/ground plane using the principal axes of objects resting on it (if present). Then, we calculate the angles between the principal axes of different objects to annotate a list of qualitative and quantitative QA pairs. We provide a subset of the question template below.

Qualitative questions:

```
Is [A] tilted?
Is [A] tilted to the left?
Is [A] inclined to the back?
Is [A] more tilted than [B]?
Is [A] more tilted than [B] to the back?
Is [A] more inclined than [B] to the right?
Is [A] leaning towards [B] vertically?
Is [A] straighter than [B]?
Along which axis (W, D, H) is [A] more tilted?
Which object(s) are not upright?
How many object(s) are not upright?
```

Quantitative questions:

```
What is the inclination angle of [A] along the vertical axis?
How many degrees is [A] tilted horizontally?
Calculate the angle of tilt for [A] towards back.
```

```
Measure the tilt of [A] relative to the front.
What is the relative angle between [A] and [B]?
Measure the angle between [A] and [B].
What is the angle between the horizontal axis of [A] and [B]?
How much is [A] inclined along the depth axis compared to [B]?
Measure the inclination difference along the vertical axis for [A] and [B].
Measure the angular deviation of [A] and [B] along the vertical axis.
Determine the angular difference between the depth axes of [A] and [B].
Determine the tilt difference between [A] and [B] along the horizontal axis.
Compare the angles of tilt for [A] and [B] along the vertical axis.
```

**Inter-Image Spatial Dynamics VQA (IrSD-VQA)** For annotation, we manually measure the changes in objects' locations and angles between two photos. To measure the change in camera shot angle, we record the change in the angle of the tripod to which the iPhone 12 Pro Max is attached. We provide a subset of the question template below.

Qualitative questions:

```
Does [A] move?
Does [A] rotate?
Does [A] rotate clockwise?
Does [A] incline?
Does [A] move to the right?
Does [A] move closer to [B]?
Does [A] become more upright?
Does [A] incline more to the back?
Does the angle between [A] and [B] become smaller?
Along which direction does [A] move?
Along which axis (W, D, H) does [A] rotate?
Which object(s) move?
How many object(s) rotate?
How many object(s) become more tilted to the back?
Does the camera shot angle change?
Along which axis (W, D, H) does the camera shot angle change?
```

Quantitative questions:

```
How far does [A] move vertically?
How far does [A] move horizontally?
How far does [A] move towards the back?
How many degrees does [A] rotate clockwise?
How many degrees does [A] rotate counterclockwise?
What is the total distance [A] moves from its original position?
Calculate the angle of inclination of [A] in the second image.
Measure the tilt of [A] relative to the first image.
What is the change in height of [A] from the first to the second image?
How much does the distance between [A] and [B] change?
How much does [A] incline towards the left compared to the first image?
What is the angular displacement of [A] towards the right?
Measure the rotation angle of [A] about its own axis.
What is the new distance between [A] and [B] in the second image?
Calculate the difference in the inclination angle of [A] between the two images.
Determine the change in angle of [A] relative to the ground plane.
What is the relative movement of [A] with respect to [B]?
Measure the angular deviation of [A] and [B] along the vertical axis.
By how many degrees has the camera shot angle changed?
Along which axis/axes has the camera shot angle changed?
How does [A] appear to move if we do not account for the camera shot angle change?
What is the perceived change in orientation of [A] due to the camera angle change?
```

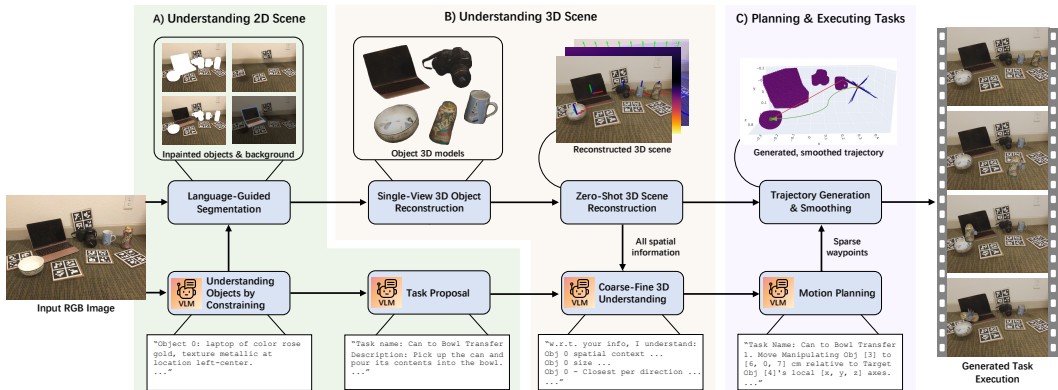

Figure 7: **Pipeline for discovering and planning for robotics tasks from a single image.** It incorporate the task proposal and motion planning modules based on SpatialPIN.

```
How does the position of [A] change relative to the camera angle difference?
What is the actual move distance of [A] when accounting for the camera shot angle change?
```

**Dataset Examples**  We illustrate example input images of all forms of spatial VQA in Fig. 6.

## C.2   Discovering and Planning for Robotics Tasks from a Single Image

Given a single RGB image of a scene with unknown environments and objects, the VLM identifies potential tasks and plans their execution using full 3D trajectories, complete with visualization. Figure 7 shows the pipeline, which incorporates task proposal and axes-constrained motion planning modules into SpatialPIN's pipeline from our main paper (Fig. 2).

**Task Proposal**  We query VLM to propose meaningful, diverse tasks, each with a one-sentence task description. Instead of directly querying VLM for task proposal, we employ a hybrid approach that integrates role-play and object-based initialization. In the role-play scenario, we prompt VLM to envision itself as a robotic/human hand working in the scene to perform household tasks. For the object-based initialization, we guide VLM to sequentially focus on each identified object within the scene. When the scene contains more than one identified objects, VLM is instructed to suggest two tasks emphasizing interactions between the manipulating object and any of the detected objects, and an additional task focused solely on the manipulating object. If only one object is detected, VLM is directed to propose a task involving just that object. This strategy guarantees a broad spectrum of task suggestions, ensuring comprehensive object engagement.

To further tailor the task proposals, we impose specific constraints, directing VLM to consider the practical affordances of objects while encouraging creative assumptions (*e.g.*, a bowl's capacity to hold water) and potential interactions (*e.g.*, transferring water from a cup into a bowl). Additionally, we delineate clear boundaries by excluding tasks that entail the disassembly of objects, functionality tests, or the involvement of imaginary objects, thereby focusing on feasible and meaningful tasks.

As a concrete example, given the image on the left of Fig. 2, with the manipulating object to be the red can, VLM will propose the following tasks:

```
''Task name:  Can to Bowl Transfer
Description:  Pick up the can and pour its contents into the bowl.
Task name:  Can Relocation
Description:  Pick up the can and place it inside the bowl.
Task name:  Can Rotation
Description:  Rotate the can 90 degrees on its vertical axis.  ...''
```

**Axes-Constrained Motion Planning through Waypoint**  We introduce a novel method to guide VLM to conduct motion planning within a 3D scene based on a proposed task by planning motion waypoints along the manipulating object's principal axes. More specifically, we define four types of manipulations that VLM can use:

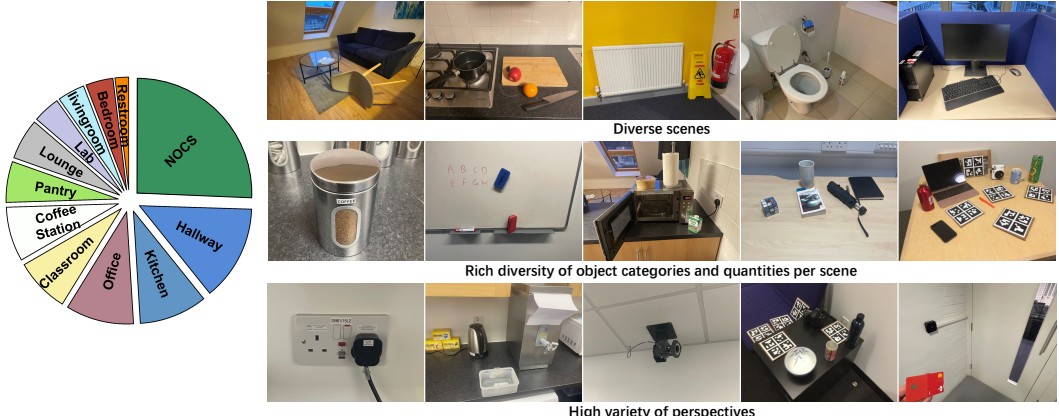

**Diverse scenes**

**Rich diversity of object categories and quantities per scene**

**High variety of perspectives**

Figure 8: **Dataset statistics.** Our dataset presents 51 scenes—13 from NOCS and 38 captured from varied perspectives—featuring a wide range of object categories, quantities, and a diverse set of tasks and planned trajectories.

*Rotation type 1:* Axial rotation. The object rotates around its principle axes.

*Rotation type 2:* Rotation relative to the target object.
- pitch: Tilt similar to pouring water, around a horizontal axis formed by the cross product of the connecting directional vector and the target's vertical axis.
- yaw: Horizontal rotation, like a camera panning, around a vertical axis formed by the cross product of the connecting directional vector and the pitch axis.
- roll: Rotation like drilling a surface, around the connecting directional vector.

*Translation type 1:* Defines the goal relative to the target object's principle axes, with translation values for its $[x, y, z]$ axes in centimeters. $[0, 0, 0]$ cm indicates the goal is the center of the target object.

*Translation type 2:* Sets the goal relative to a directional vector between two reference objects, specifying how far (in cm) object 1 should move towards or away from object 2 along this vector.

Since VLM inherently lacks the capability to provide 3D coordinates and low-level actions directly [60, 19], our method offers a practical workaround by translating natural language instructions into precise motion waypoints. This approach significantly enhances VLM's utility in spatial reasoning and manipulation tasks without requiring direct 3D coordinate generation capabilities. Also, the four types of manipulations we defined are both simple and comprehensive, covering a broad spectrum of manipulation tasks.

As a concrete example, given the image on the left of Fig. 7, with ``Task name: Can to Bowl Transfer'', VLM will plan as follows:

```
``Task Name: Can to Bowl Transfer
Manipulating obj idx: 3
Interacting obj idx: 4
1. Move Manipulating Obj [3] to [6, 0, 7] cm relative to Target Obj [4]'s local [x, y, z] axes.
2. rotate_wref: Rotate Manipulating Obj [3] relative to Target Obj [4] around [pitch] axis by [75]
degrees.''
```

In practice, the quality of motion planning by VLMs can be enhanced using various prompting techniques. One such technique is chain-of-thought (CoT) [63], where another LLM guides the VLM to plan each axes-constrained sparse waypoint step by step.

**Dataset Statistics** See Fig. 8 for statistics and visuals. We standardize all image dimensions by resizing all to $640 \times 480$.

**More Qualitative Demonstration** We present more qualitative examples spanning various scenes and tasks, as shown in Fig. 9.

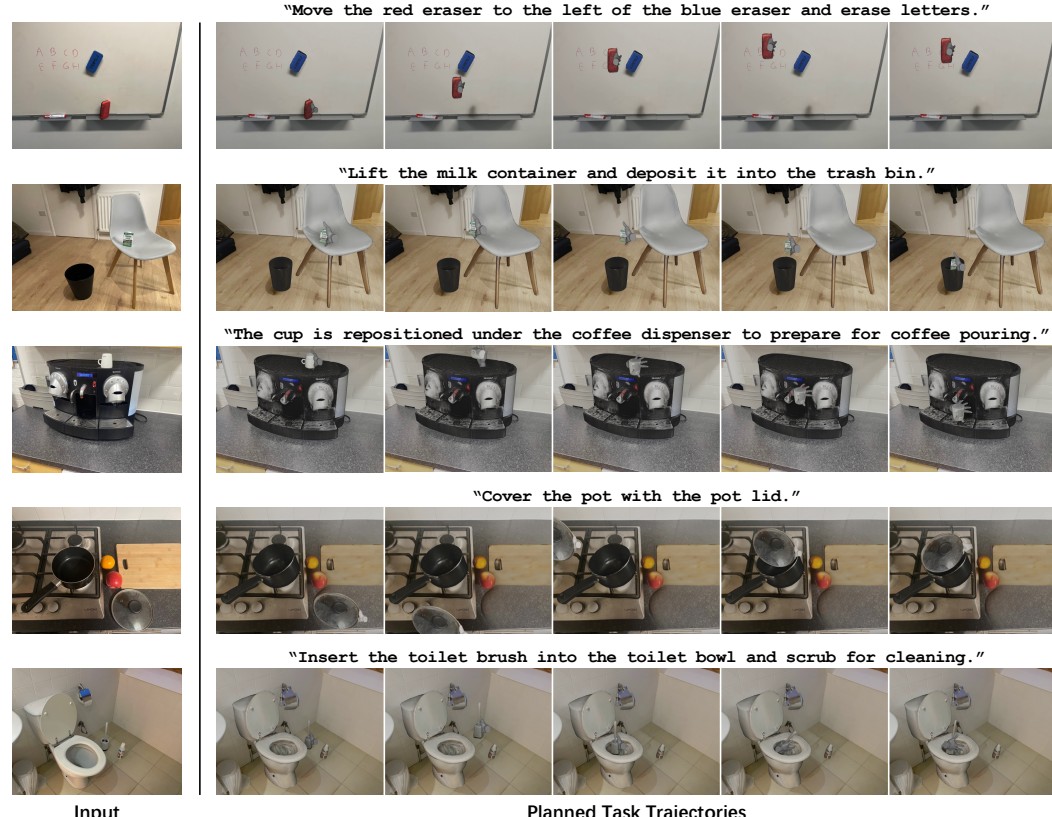

| Input | Planned Task Trajectories |

Figure 9: **More qualitative examples.** With diverse input scenes and proposed tasks, our framework produces 3D trajectories with geometric awareness that aligns with the task descriptions. *Zoom in for better view.*

Table 10: **Comparison of task diversity.** We sample 106 proposed tasks for fair comparison with RoboGen and previous RL benchmarks.

|  | **Ours** | RoboGen [61] | Behavior-100 [54] | RLbench [28] | MetaWorld [72] | Maniskill2 [23] |
|---|---|---|---|---|---|---|
| Number of Tasks | 106 | 106 | 100 | 106 | 50 | 20 |
| Self-BLEU ↓ | **0.269** | 0.284 | 0.299 | 0.317 | 0.322 | 0.674 |
| Embedding Similarity ↓ | **0.154** | 0.165 | 0.210 | 0.200 | 0.263 | 0.194 |

## Experiment on Human Understanding

We aim to understand human perception of our generated task executions by asking participants to provide a one-sentence description. We then evaluate the alignment between these descriptions and the ground truth task descriptions proposed by VLM, using OpenCLIP [15, 43]. Table. 9 reveals a high degree of alignment. Intriguingly, it appears humans understand our task executions more accurately than machines do (Table. 7). We hypothesize this discrepancy stems from the limitations of current video understanding models, whereas humans draw on their prior experiences for a deeper comprehension.

Table 9: Results for machine understanding (generation) on 278 task executions and human understanding, where 25 users write descriptions for 10 tasks.

|  | **Machine** | **Human** |
|---|---|---|
| OpenCLIP ↑ | 0.636 | 0.823 |

**Experiment on Task Diversity** We evaluate the diversity of the proposed tasks in terms of semantic meaning using Self-BLEU and the embedding similarity [75, 48, 49] following RoboGen [61], where lower scores mean higher diversity. We also compare with previous reinforcement learning (RL) benchmarks. From Table. 10, ours method generates most diverse tasks as it is open to all scenes with no constraint.

## C.3    Trajectory Generation Implementation Details

**Trajectory Generation**   With the waypoints planned by VLM, we generate the manipulating object's trajectory using path planning algorithm, specifically rapidly-exploring random tree star (RRT*) [31]. To generate accurate collision-free path, we perform K-means clustering on the point clouds of object 3D model with a high number of clusters, segmenting the object mesh into discrete voxels and treating each voxel as an obstacle. Then, to accurately consider the manipulating object's dimensions, we grow the size of each voxel by its dimensions.

**Handling VLM Planning Discrepancies**   The waypoints generated by VLM are typically accurate and practical. Nonetheless, there are instances where the waypoints suggested by VLM lead to collisions as determined by the RRT* planner. This discrepancy is less about VLM's misunderstanding of the objects' sizes and their spatial relationship and more about the precision level of the waypoints, which may not match the exacting standards of the RRT* planner's outcomes. To resolve this, we implement Gaussian sampling around the initially planned waypoints whenever a collision is detected. The sampling strategy is guided by a predefined set of geometric rules. In our 3D coordinate system, positive x-axis [1, 0, 0] points right, positive y-axis [0, 1, 0] is away from viewer, positive z-axis [0, 0, 1] is up. For translation type 1, we denote the goal pose relative to the target object's principle axes as [dx, dy, dz]. For translation type 2, we denote the distance that object 1 moves towards object 2 as dD. The set of geometric rules are as follows:

if type 1 $\&$ [dx, dy, dz] = [0, 0, 0]: sample along [x, y, z] axes
elif type 1 $\&$ dx = 0 $\&$ dy = 0 $\&$ dz != 0: sample along [z] axis, $z_{sampled} \cdot dz > 1$
elif type 1 $\&$ dz = 0: sample along [x, y] axes
elif type 1 $\&$ dz != 0: sample along [x, y, z] axes, $z_{sampled} \cdot dz > 1$

if type 2: sample along the connecting directional vector, $D_{sampled} \cdot dD > 1$

**Trajectory Smoothing**   Finally, to ensure our trajectory is natural and smooth, we linearly interpolate rotation and interpolate translation using cubic spline.

# D    Prompt Details

We show exact prompts for VLMs for our proposed application: discovering and planning for robotics tasks from a single image.

## Understanding Objects by Constraining Prompt.

```
Input:
RGB image (640, 480) = (width, height) with multiple objects.

Your task is to identify and objects by precise color, texture, and 2D spatial locations (in words).
Do not use vague phrase like multi-colored.

Please write in the following format.  Do not output anything else:
Object idx (actual integer, start from 0):  x of color y, texture z at location w.
```

## Task Proposal Prompt.

```
Input:
1.  RGB image (640, 480) = (width, height) with multiple objects.
2.  Detected objects with index.

You are a single robot hand working in this image scene to perform simple household tasks.  Tasks must
be discovered from the image.  Consider objects' affordances and feel free to make assumptions (e.g., a
bowl can contain water) and interactions with other objects (e.g., pouring water from a cup into a bowl).

Task types:
1.  Interaction between the manipulating object and one of the detected objects (involve translation, or
```

```
translation + rotation).
2.  Rotate manipulating object (involve rotation).

Strictly follow constraints:
1.  Exclude tasks involving disassembly of objects.
2.  Exclude tasks involving cleaning or functionality testing.
3.  Exclude tasks involving imaginary objects.
4.  Manipulating object moves; interacting object static.
5.  Assume all objects are rigid, without joints or moveable parts (i.e., cannot deform, disassemble,
transform).  This applies even to objects that are typically articulated (e.g., laptop).

Propose 3 tasks (2 interaction, 1 rotation) for manipulating Object 5.  Write in the following format.
Do not output anything else:
Task Name:  xxx
Manipulating obj idx:  5
Interacting obj idx:  obj_idx (actual integer, or manipulating obj idx)
Description:  basic descriptions.
```

## Coarse 3D Understanding Prompt.

```
Inputs:
1.  RGB image (640, 480) = (width, height) with multiple objects
2.  Detected objects with index.
3.  Image scene size.
4.  Maximum and minimum width, depth, and height.

Your task is to identify the camera shot angle (horizontal, top-down, bottom-up).  Reason with respect to
the visual cues, the image scene size, and maximums and minimums along each dimension.  Choose horizontal
if not severely angled.

Please write in the following format.  Be concise.  Do not output anything else:
Visual cues reasoning:  ...
Spatial data reasoning:  ...
Conclusion:  horizontal/top-down/bottom-up.
```

## Image and Spatial Context Understanding Prompt.

```
Inputs:
1.  RGB image (640, 480) = (width, height) with multiple objects and their visualized local axes (x red, y
green, z blue).
2.  Detected objects with index.
3.  For each detected object, its 3D center, local xyz-axes, size, and spatial relationship relative to
other objects.

The 3D coordinate system of the image is in centimeters and follows Blender.  Positive x-axis [1, 0,
0] right, positive y-axis [0, 1, 0] away from viewer, positive z-axis [0, 0, 1] up.  Positive rotation is
counter-clockwise around all axes.

Your task is to learn the spatial context.  Do not output.
```

## Motion Planning Prompt.

```
Inputs:
1.  RGB image (640, 480) = (width, height) with multiple objects and their visualized local axes (x red, y
green, z blue).
2.  Detected objects with index.
3.  Simple household tasks and descriptions to be performed by a single robot hand.
```

Your goal is to plan fine-grained motions for the manipulating object to complete the tasks using four manipulations, explained as follows:

Rotation:
rotate_self: Axial rotation. The object rotates around its local [x/y/z] axis by [degrees].
rotate_wref: Rotation relative to the target object:
- pitch: Tilt similar to pouring water, around a horizontal axis formed by the cross product of the connecting directional vector and the target's z-axis.
- yaw: Horizontal rotation, like a camera panning, around a vertical axis formed by the cross product of the connecting directional vector and the pitch axis.
- roll: Rotation like a drill entering a surface, around the connecting directional vector.
The degrees can be specified in two ways:
- Exact [degrees]. Positive values rotate the manipulating object towards the target object.
- Fixed_towards/fixed_back. 'fixed_towards' orients the object towards the target, mimicking actions like pouring (pitch), facing (yaw), or drilling into (yaw+roll) the target. 'fixed_back' reverses this alignment.

Translation:
translate_tar_obj: Defines the goal relative to the target object's local axes, with translation values for its [local_x, local_y, local_z] axes in centimeters. [0, 0, 0] cm indicates the goal is the center of the target object.
translate_direc_axis: Sets the goal relative to a directional vector between two reference objects, specifying how far (in cm) object 1 should move towards or away from object 2 along this vector (positive closer, negative away). Object indices must differ, and if one reference object is the manipulating object, its current location is used.

Strictly follow caveats:
1. Apply rotate_wref thoughtfully and sequentially around different axes as needed.
2. Use the provided spatial information and image effectively for understanding and planning within the 3D scene.
3. Combine common physical understanding with the scene's spatial details (like relative positions and sizes of objects) for strategic planning.
4. Remember that objects' local axes' positive directions might require using negative values in rotation and translation for authentic motion planning.

Plan as below. Fill in obj_idx based on the tasks.
rotate_self: Rotate Manipulating Object [obj_idx] around its local axis [x/y/z] by [degrees].
rotate_wref: Rotate Manipulating Object [obj_idx] relative to Target Object [target_obj_idx] around [pitch/yaw/roll] axis by [degrees/fixed_towards/fixed_back].
translate_tar_obj: Move Manipulating Object [obj_idx] to [a, b, c] cm relative to Target Object [target_obj_idx]'s local [x, y, z] axes.
translate_direc_axis: Move Manipulating Object [obj_idx] [a] cm along the directional vector from Reference Object [ref_obj_1_idx] to Reference Object [ref_obj_2_idx].

Here are some full examples. Please write in the following format. Do not output anything else:
Task Category: Bear rotation
Description: Rotate the toy bear 90 degrees on its vertical axis.
Motion Planning:
Manipulating obj idx: bear_idx (actual integer)
Interacting obj idx: bear_idx (actual integer)
1. rotate_self: Rotate Manipulating Object [bear_idx] around its local axis [z] by [90] degrees.

Task Name: Cup content transfer
Description: Pick up the mug and pour its contents into the bowl.
Motion Planning:
Manipulating obj idx: cup_idx (actual integer)
Interacting obj idx: bowl_idx (actual integer)
1. translate_tar_obj: Move Manipulating Object [cup_idx] to [5, -7, 5] cm relative to Target Object [bowl_idx]'s local [x, y, z] axes.

2. rotate_wref: Rotate Manipulating Object [obj_idx] relative to Target Object [bowl_obj_idx] around [pitch] axis by [fixed_towards].

Task Name: Screwdriver penetration
Description: Use a screwdriver to penetrate an avocado.
Motion Planning:
Manipulating obj idx: screw_idx (actual integer)
Interacting obj idx: avocado_idx (actual integer)
1. translate_tar_obj: Move Manipulating Object [screw_idx] to [-5, -5, 0] cm relative to Target Object [avocado_idx]'s local [x, y, z] axes.
2. rotate_wref: Rotate Manipulating Object [screw_idx] relative to Target Object [avocado_idx] around [yaw] axis by [fixed_towards].
3. rotate_wref: Rotate Manipulating Object [screw_idx] relative to Target Object [avocado_idx] around [roll] axis by [360] degrees.

