# OpenReview forum: "SpatialPIN: Enhancing Spatial Reasoning Capabilities of Vision-Language Models through Prompting and Interacting 3D Priors"
_NeurIPS.cc/2024/Conference — NeurIPS 2024 poster_

### Official Review · Reviewer_C3yx · 2024-07-11

**Soundness:** 3
**Presentation:** 3
**Contribution:** 3
**Rating:** 6
**Confidence:** 4

**Summary:**

This paper introduces SpatialPIN to enhance the spatial reasoning capabilities of Visual Language Models (VLMs) by explicitly incorporating 3D priors from 3D foundation models. Extensive experiments demonstrate that SpatialPIN is effective for spatial Visual Question Answering (VQA) and robotic pick-and-place tasks.

**Strengths:**

1.	The method is plug-and-play and can enhance the spatial reasoning capabilities of VLMs without requiring additional training.
2.	Experiments are comprehensive, validating the approach on both spatial VQA and robotic tasks.

**Weaknesses:**

1.	The writing in the methodology section needs improvement. Sections 3.1 and 3.2 mix method descriptions with prompts and corner cases, making it challenging for readers to understand.
2.	I'm uncertain about the extent to which fine-grained spatial relationships benefit robots. More experiments are needed, particularly beyond simple pick-and-stack tasks. Additionally, validation in real-world scenarios, combining camera matrices and depth estimation, would be beneficial.

**Questions:**

1.	Can the proposed method be compared with affordance-based methods like Voxposer? What are the advantages and disadvantages?

**Limitations:**

Refer to weaknesses.

---

> ### Author Rebuttal · Authors · 2024-08-07
>
> **Q1**: Sections 3.1 and 3.2 mix method descriptions with prompts and corner cases, making it challenging for readers to understand.
>
> **A1**:
> Thank you for pointing this out! Since our methodology involves VLMs interacting with 3D priors, the output of VLMs sometimes serves as input for 3D foundational models (2D Understanding), and sometimes VLMs use the output of 3D foundational models as input (Coarse and Fine-Grained 3D Understanding). Our original intention was to guide readers through our pipeline step by step, which is why the method descriptions interleave with the prompts.
>
> To make our Methodology section easier to understand, we will use colored panels to detach the prompts and the method descriptions. We will also put some detailed but lengthy examples of prompting into the Appendix for better coherence. Besides, we consider labeling the section number and workflow index on our pipeline in Fig. 2 and changing the naming of "Prompting: ..." to the module name of our pipeline such as "Understanding Objects by Constraining" in the subsections in our Methodology. Finally, we will make sure we clearly explain the input/output workflow in our pipeline.
>
> ---
>
> **Q2**: What is the extent to which fine-grained spatial relationships benefit robots? More experiments are needed, particularly beyond simple pick-and-stack tasks.
>
> **A2**:
> Thank you for the suggestion. We definitely agree that more robotics experiments would make our framework more comprehensive. To address your concern about the extent to which fine-grained spatial relationships benefit robots, we add a new experiment of peg insertion. Peg insertion is a task that requires a finer level of spatial reasoning and higher precision than pick-and-stack. The challenges are threefold:
> 1. VLM needs to reason about the size of the peg (how much to grasp).
> 2. VLM needs to determine which principal axis and by how much the robotic arm should rotate the peg to align its orientation with the cube.
> 3. This also evaluates the accuracy of the 3D centers of the peg, the cube, and the small hole, as well as the principal axes output by our framework.
>
> We set up the peg insertion problem in the ManiSkill simulator, applying real-world physics properties. We randomize the size and location of the peg and the cube, and the size and position of the small hole on the cube. We creat 30 scenes.
>
> We compare our method with current baselines (details in Section 4.2 of our paper) plus VoxPoser [1]. For the method using direct 3D information from our framework, we assume the robot grasper should grasp the smallest side of the peg. There is no principal axes alignment, and RRT* directly plans the trajectory from the peg to the small hole. For VoxPoser, we set up the peg insertion problem in RLBench, as their code supports this scenario. Using the same approach as above, we create 30 scenes. Results are shown in **Table. T5** and the demo in **Fig. F3**. From the results, our method has higher success rates than VoxPoser. For both the direct 3D information method and the SpatialVLM [2] method, the success rates are lower since the orientation of the peg is not aligned or adjusted properly. However, there are still success cases where the randomized orientation of the peg happens to be aligned with the cube at the beginning.
>
> At the same time, we would like to highlight that we included 8 experiments, ranging from different forms of VQAs and robotics tasks to coming up with potential trajectories given an input image, without counting ablation studies. The number of experiments is on par with, if not more than, many existing works such as SpatialVLM [2], published in top venues. SpatialVLM has 2 experiments and 2 ablations for VQAs, and 1 experiment for robotics.
>
> ---
>
> **Q3**: Can the proposed method be compared with affordance-based methods like VoxPoser? What are the advantages and disadvantages?
>
> **A3**:
> VoxPoser [2], similar to ours, can generate trajectories for manipulation tasks given unknown scenes in a zero-shot manner. In VoxPoser, the VLM is not used for spatial reasoning but is used to detect object affordances to generate 3D value maps, which serve as objective functions/dense rewards for motion planners. The objects of interest are granted affordance maps while other objects are granted avoidance maps. The use of affordance maps enables VoxPoser to generate more feasible grasps on some of the objects than ours (e.g., grasp the handle of a bag), but we consider this part can be enhanced by a simple extension to our pipeline (e.g., using VLM to detect affordances) in our 2D Scene Understanding stage.
>
> However, VoxPoser's generated 3D value maps are coarse as a smoothing operation is applied to them, making it challenging for the motion planner to handle scenes/tasks involving a large number of objects or spatial tasks that require high precision regarding objects' shapes and poses (e.g., peg insertion). Please see **Table. T5** for results and **Fig. F3** for demo. Instead, our method injects explicit 3D scene understanding information into VLMs through progressive prompting, enabling VLMs to plan for tasks that require a finer level of spatial understanding such as precise peg insertion. Therefore, as shown in **Table. T5**, our method outperforms VoxPoser by a fair margin in the peg insertion task.
>
> ---
>
> **References:**
>
> [1] Huang, Wenlong, et al. "VoxPoser: Composable 3D Value Maps for Robotic Manipulation with Language Models." CoRL, 2023.
>
> [2] Chen, Boyuan, et al. "Spatialvlm: Endowing vision-language models with spatial reasoning capabilities." CVPR, 2024.

---

> > ### Comment · Reviewer_C3yx · 2024-08-13
> >
> > Thank you for the insightful response. My concerns have been addressed overall, and I intend to maintain the original rating.

---

> ### Author Response · Authors · 2024-08-13
>
> We are happy that our replies address your questions! We sincerely appreciate your kind response and positive feedback.

---

### Official Review · Reviewer_K1EW · 2024-07-12

**Soundness:** 3
**Presentation:** 3
**Contribution:** 3
**Rating:** 6
**Confidence:** 3

**Summary:**

This paper presents SpatialPIN, a framework designed to enhance the spatial reasoning capabilities of Vision-Language Models (VLMs) through prompting and interacting with 3D priors from multiple foundation models in a zero-shot, training-free manner. The authors argue that current state-of-the-art spatial reasoning-enhanced VLMs, which are trained on spatial visual question answering (VQA) datasets, may not generalize well to more complex 3D-aware tasks. SpatialPIN aims to address this by incorporating explicit 3D scene understanding through progressive prompting and interactions between VLMs and 2D/3D foundation models. The framework is evaluated on various spatial reasoning tasks, including different forms of spatial VQA and robotics applications like pick-and-stack and trajectory planning.

**Strengths:**

1. The approach of combining VLMs with 3D foundation models for spatial reasoning is novel and addresses the limitations of current methods that rely solely on training on spatial VQA datasets.
2. The paper is generally well-written and organized.
3. The work addresses an important gap in current VLMs' spatial reasoning capabilities, which has implications for various applications, particularly in robotics.

**Weaknesses:**

1. While the integration of 3D priors is novel, the combination of techniques used in the framework may not be entirely new, as it builds on existing 3D and VLM methodologies.
2. The reliance on multiple 3D foundation models might introduce complexities and dependencies that are not fully addressed in terms of scalability and robustness.
3. The use of proprietary VLMs as the core renders the proposed pipeline less insightful as the proprietary VLMs' capability of utilizing 3D information is a mystery and the performances are hard to analyze.

**Questions:**

1. Can the authors provide more details on the inference time for the full pipeline? How might this be optimized for real-time applications?

2. How sensitive is the performance to the quality of the 3D reconstruction proposed in Section 3.3.? What happens when the reconstruction is imperfect or fails and how do the authors handle this issue?

3. What do the authors mean by "the VLM discovers potential tasks"? Is it to allow the VLM to come up with feasible tasks given the objects in the scene and then to solve the self-proposed tasks? If so

**Limitations:**

The paper acknowledges the limitations related to inference speed and dependency on the quality of 3D foundation models. However, it would be constructive to discuss potential solutions or future directions to address these limitations.

---

> ### Author Rebuttal · Authors · 2024-08-06
>
> **Q1**: The combination of techniques used in SpatialPIN may not be entirely new, as it builds on existing 3D and VLM methodologies.
>
> **A1**: While using existing 3D and VLM methods, composing the framework of SpatialPIN isn't trivial and requires many thoughtful designs to first understand 2D objects, incorporate coarse 3D understanding, and inject fine-grained 3D info. Additional details such as background inpainting before depth estimation, and perspective canonicalization also help in determining many ambiguities during inference.
>
> ---
>
> **Q2**: The reliance on multiple 3D foundation models may introduce complexities and dependencies that are not fully addressed in terms of scalability and robustness.
>
> **A2**: We agree that accumulated errors occur when each foundational model depends on previous output. In particular, we notice cases where imperfect 3D reconstruction leads to suboptimal results. Additional explanations and statistics are provided in response to Q5.
>
> Meanwhile, we would like to highlight that the current robustness of 6D pose and size estimation (detailed in Section 3.3 & Appendix B) is greatly helpful in the recovery of the 3D spatial space even with suboptimal object reconstruction (examples in **Fig. F2**). Based on all experiments in the paper, the accumulated errors on multiple datasets are significantly lower than our baselines such as SpatialVLM. We will discuss this further in the Limitations of our paper.
>
> ---
>
> **Q3**: The use of proprietary VLMs as the core renders the proposed pipeline less insightful as the proprietary VLMs' capability of utilizing 3D information is a mystery and the performances are hard to analyze.
>
> **A3**: Thank you for the insight! We definitely agree that using proprietary VLMs may cause difficulties in analyzing some of the performances. We follow numerous prior notable works (FoundationPose [1], VoxPoser [2]) and use GPT-4 as it brings the best performances.
>
> To better evaluate SpatialPIN, we provide additional experiments with the newly released LLaMa 3.1-70B as our backbone on the task of IaOR-VQA as shown in **Table. T2** & **T3**.
>
> ---
>
> **Q4**: More details on the inference time for the full pipeline and optimization for real-time applications.
>
> **A4**: We provide detailed inference speeds for 30 input images with varying object categories and quantities (1-7) on the task of IaOR-VQA. For the 3D object reconstruction process, we vary the resolution of the input images (higher resolution generally leads to better quality but longer inference time). As detailed in Appendix C.2, we standardize all input images to 640 × 480 for an optimal trade-off between quality and inference speed. For VLMs, we compare API calls to GPT-4o (T3 personal account) and the open-source VLM, LLaVA-1.5.
>
> In our implementation, we optimize the inference speed by running coarse and fine-grained 3D understanding in parallel. Additionally, we apply several implementation tricks to reduce the inference time of the 3D reconstruction process:
> 1. We use One-2-3-45++, a two-stage pipeline where the first stage outputs 3D model shapes/normals and the second stage outputs colors/textures. For VQA and robotic planning tasks, we only need the output from the first stage, which reduces inference time by ~85% (completing two stages takes ~55s).
> 2. We reconstruct each object in parallel using a combination of local inference and API calls, as One-2-3-45/One-2-3-45++ [3, 4] is commercialized.
>
> From **Table. T4**, we can see that the 3D reconstruction is the bottleneck. Yet, we highlight that:
> 1. This process runs only once per image. If the user asks a second question for the same image, the inference time of our entire framework equals the VLM inference time.
> 2. The speed is expected to improve with future versions of 3D foundation models. For example, LRM [5], a recent single-view 3D reconstruction model, reconstructs an object within 5s.
>
> ---
>
> **Q5**: Sensitivity of the performance to the quality of 3D reconstruction and handling imperfect/failed reconstruction.
>
> **A5**: Single-view 3D reconstruction is a challenging ill-posed problem, especially given the complexity of our datasets with diverse scenes, object categories, quantities, and camera angles. To tackle this, we refine the 2D representation for each object by inpainting to handle occlusions (Section 3.1).
>
> In most cases, our pipeline results in high-quality reconstruction. For visual comparisons of reconstructed scenes, please refer to PDF **Fig. F1** and  Appendix Fig. 9. However, imperfect reconstruction cases can still occur. Our statistics indicate that in a set of 100 images containing 1-7 objects, ~6 images have imperfectly reconstructed objects. However, our experiments demonstrate that even in these cases, the spatial reasoning-enhanced VLMs generally perform well in VQAs due to the robustness of our 6D pose and size estimation. Examples of such imperfect reconstruction cases are also provided in **Fig. F2**.
>
> ---
>
> **Q6**: Meaning of "VLM discovers potential tasks".
>
> **A6**: Given a single RGB image of a scene, we query the VLM to propose meaningful, diverse tasks, each with a one-sentence description. Instead of directly querying the VLM for task proposals, we use a hybrid approach that integrates role-play (envision itself as a robotic/human hand) and object-based initialization (guide it to sequentially focus on each identified object). Please refer to Appendix C.2 for method details and Appendix D for prompt.
>
> ---
>
> Reference:
>
> [1] Foundationpose: Unified 6d pose estimation and tracking of novel objects. CVPR 2024
>
> [2] VoxPoser: Composable 3D Value Maps for Robotic Manipulation with Language Models. CoRL 2023
>
> [3] One-2-3-45++: Fast single image to 3d objects with consistent multi-view generation and 3d diffusion. CVPR 2024
>
> [4] One-2-3-45: Any single image to 3d mesh in 45 seconds without per-shape optimization. NeurIPS 2023
>
> [5] LRM: Large Reconstruction Model for Single Image to 3D. ICLR 2024

---

> > ### Comment · Reviewer_K1EW · 2024-08-13
> > **Response to the authors' rebuttal**
> >
> > Thank you for providing the detailed answers. My concerns have been mostly resolved. The rating of 6 is maintained.

---

> ### Author Response · Authors · 2024-08-13
>
> Thank you for maintaining a positive rating and we are happy to have addressed your concerns!

---

### Official Review · Reviewer_gF9G · 2024-07-12

**Soundness:** 3
**Presentation:** 2
**Contribution:** 2
**Rating:** 4
**Confidence:** 3

**Summary:**

This paper presents a pipeline designed to equip 2D Vision Language Models  with the capability to understand 3D spatial relationships. The key benefits of this framework are its zero-shot, training-free nature. The effectiveness of this approach has been validated through experiments on spatial Visual Question Answering (VQA) and various robotics tasks.

**Strengths:**

1. Prompting VLM model for spatial awareness is an important field. The author provides a feasible solution.

**Weaknesses:**

1. The motivation behind this paper appears ambiguous. The authors assert that "high-level 3D-aware tasks are underexplored," yet there is notable prior work such as "Spatial VLM: Endowing Vision-Language Models with Spatial Reasoning Capabilities" and a series of studies involving GPT-4-V for robotics that address high-level 3D-aware tasks. It may be beneficial for the authors to refine the principal motivation behind SpatialPIN to distinguish it more clearly from existing research.
2. The distinction between SpatialPIN and other forms of spatial prompting is not clearly articulated in the introduction. This section could benefit from a more detailed comparison to enhance clarity and strengthen the justification for SpatialPIN's unique contributions.
3. The claim that operating without fine-tuning represents a contribution seems questionable, as existing works involving LLM agents and LLM prompting also employ techniques that do not require tuning. Clarification on why this aspect is particularly innovative or advantageous in the context of SpatialPIN would be helpful.
4. There is room for improvement in the writing, including the presentation in Figure 1, the text , and the sections detailing the main contributions.

**Questions:**

See Weaknesses.

**Limitations:**

Yes.

---

> ### Author Rebuttal · Authors · 2024-08-07
>
> **Q1**: Ambiguity in motivation. The authors assert that "high-level 3D-aware tasks are underexplored," yet there is notable prior work such as SpatialVLM and a series of studies involving GPT-4V for robotics that address high-level 3D-aware tasks.
>
> **A1**: We thank the reviewer for highlighting this issue and hope the following response addresses the ambiguity.
>
> To clarify, we define high-level 3D-aware tasks as tasks that consider the semantics of the objects to provide more accurate predictions in the spatial relationships (e.g., measurements between two objects) and potential plans for executing robotic tasks (e.g., pick and stack, task trajectory planning).
>
> SpatialVLM [1] is a powerful and recent model that enhances the spatial capabilities of VLMs by finetuning them on spatial VQA datasets. However, since most of the spatial VQA questions are on surface-level distance relationships (how far apart are object A and B), we argue that the semantics of objects are not well considered in the VLM compared to explicit 3D information, which SpatialPIN leverages. This is supported by several of our experiments, including Fig. 4 in our paper which shows that SpatialPIN provides much more accurate answers on more diverse and challenging forms of VQAs (Tables. 1-4) as it explicitly considers the semantic/geometry of the objects (knowing how big/tilted a laptop and a mug is by reconstruction first), as well as Table. 5 where explicit 3D information enables VLMs to act as a strong prior for tasks like pick and stack which SpatialVLM was less successful at. We will refine these descriptions and add a detailed motivation in our Introduction.
>
> Meanwhile, we agree with the reviewer that addressing several robotics 3D-aware tasks utilizing GPT-4V is beneficial to our work and we list their differences as below:
>
> 1. VLM as Success Detectors [2]. This work uses VLMs to output yes/no answers to determine if the robotics pick-and-place task is completed based on RGB images but cannot perform other downstream tasks.
>
> 2. VLMs are Zero-Shot Reward Models for RL [3]. This work uses VLMs' trained encoder as a CLIP model to output rewards given observations in simulation/gaming environments on tasks that do not require 3D spatial understanding.
>
> 3. RoboGen [4]. This work uses VLMs to set up scenes in the simulator using its common sense. For example, VLMs output that the oven should be put on the table. These understandings do not require detailed spatial relationships nor task planning.
>
> 4. VoxPoser [5]. The VLM is not used for spatial reasoning but used to detect object affordances to generate 3D value maps, which serve as objective functions/dense rewards for motion planners. However, the generated 3D value maps are coarse as a smoothing operation is applied to them, making it challenging for the motion planner to handle scenes/tasks involving a large number of objects or spatial tasks that require high precision regarding objects' shapes and poses (e.g., peg insertion). We provide an additional experiment on peg insertion in **attached pdf Table. T5** and demo in **Fig. F3**, comparing our performance with VoxPoser. We set up the peg insertion problem in 30 scenes using the ManiSkill simulator applying real-world physics properties. More details can be found in our reply to Reviewer C3yx.
>
> ---
>
> **Q2**: The distinction between SpatialPIN and other forms of spatial prompting.
>
> **A2**:
> Thank you for pointing this out! There are various prompting methods to improve the output quality of LLMs on general VQA, such as Chain-of-Thought (CoT), Tree-of-Thought (ToT), etc. However, enhancing VLM's 3D spatial reasoning capabilities through prompting without training is still an underexplored area. To the best of our knowledge, we are the first to investigate how to inject explicit 3D scene understanding into VLMs through progressive prompting and interactions between VLMs and 2D/3D foundation models. As supported by Reviewer K1EW, our meticulously designed progressive prompting and interactions between VLMs and 2D/3D foundation models is a novel approach.
>
> There are some existing works that also utilize the idea of prompting 3D information into VLMs, such as Chat-3D [6], ShapeLLM [7], and Agent3D-zero [8]. However, they all require 3D scans as inputs and have no way to interpret the semantic meanings of single images. SpatialVLM is the closest in terms of the inputs and tasks that we can perform but is empirically shown by our results to perform less well on a variety of tasks.
>
> Finally, to further provide a more comprehensive study against other formats of spatial prompting, we also amend SpatialPIN to take in 2D screenshots of 3D scans directly and compare it on ScanQA dataset with Chat3D and Chat3D-v2 [8], as shown in **attached PDF Table. T1**. Our results indicate superior performance even with the visual imperfections of 2D screenshots of 3D scans.
>
> ---
>
> **Q3**: The claim that operating without finetuning represents a contribution seems questionable.
>
> **A3**: While many LLM prompting techniques are training-free, SpatialPIN's plug-and-play module, as supported by Reviewer C3yx, has the advantage that future improvements on individual foundation models (e.g., 3D reconstruction) can directly replace certain components in this framework for better performance. As there currently exists no comprehensive 3D-VQA dataset that tackles all the tasks SpatialPIN can currently perform (spatial relationships, pick and stack, coming up with potential tasks and trajectories), we argue that a training-free approach is a contribution to the community in chaining 3D foundation models for a diverse set of tasks.
>
> Overall, given the strong results and being the first to combine explicit 3D predictions with VLM prompting for various downstream tasks, we strongly believe SpatialPIN's acceptance would be beneficial to the vision community.
>
> ---
> **References:**
>
> We put it in the Official Comment due to word limit.

---

> ### Author Response · Authors · 2024-08-07
> **References**
>
> **References:**
>
> [1] Chen, Boyuan, et al. "Spatialvlm: Endowing vision-language models with spatial reasoning capabilities." CVPR, 2024.
>
> [2] Du, Yuqing, et al. "Vision-Language Models as Success Detectors." CoLLAs, 2023.
>
> [3] Rocamonde, Juan, et al. "Vision-Language Models are Zero-Shot Reward Models for Reinforcement Learning." ICLR, 2024.
>
> [4] Wang, Yufei, et al. "RoboGen: Towards Unleashing Infinite Data for Automated Robot Learning via Generative Simulation." ICML, 2024.
>
> [5] Huang, Wenlong, et al. "VoxPoser: Composable 3D Value Maps for Robotic Manipulation with Language Models." CoRL, 2023.
>
> [6] Wang, Zehan, et al. "Chat-3d: Data-efficiently tuning large language model for universal dialogue of 3d scenes." arXiv, 2023.
>
> [7] Qi, Zekun, et al. "Shapellm: Universal 3d object understanding for embodied interaction." arXiv, 2024.
>
> [8] Zhang, Sha, et al. "Agent3D-Zero: An Agent for Zero-shot 3D Understanding." arXiv, 2024.

---

> ### Author Response · Authors · 2024-08-13
>
> Dear Reviewer,
>
>
> We greatly appreciate your constructive feedback and concerns! We have carefully answered your questions with experiments, and sincerely hope that these have addressed your concerns. Once again, we look forward to your response after going through our rebuttal and other reviews. If there are any further issues or feedback, we are happy and eager to address them promptly.
>
>
> Thank you once again!

---

> ### Comment · Reviewer_gF9G · 2024-08-14
> **Thank the authors for the detailed response.**
>
> I have read the rebuttal and other reviews. Some of my concerns have been solved by the rebuttal, but I still feel that the technical contribution of this work is somewhat incremental and the novelty is limited. I have increased my score to "4: Borderline Reject" accordingly.

---

> ### Author Response · Authors · 2024-08-14
>
> Thank you so much for your responses and for raising our score. Your comments have been helpful in pushing the paper towards better quality.
>
> Since it is near the end of the discussion period, we cannot contribute more meaningfully to good discussions. Still, we want to highlight our contributions:
>
> 1. We investigate how to address an important gap in current VLMs' spatial reasoning capabilities. We propose SpatialPIN, a modular plug-and-play framework that progressively enhances VLM's 3D reasoning capabilities by prompting and interacting with 3D foundational models, which is the first of its kind.
>
> 2. Our approach better considers object semantics by providing VLMs with explicit 3D information. We address the limitations of current methods that rely solely on training on spatial VQA datasets, which may not generalize well to more complex 3D-aware tasks.
>
> 3. We validate our method with extensive experiments, ranging from spatial VQAs to various robotic tasks. The number of experiments to measure SpatialPIN's robustness is on par with, and even exceeds, many previous works.

---

### Author Rebuttal · Authors · 2024-08-06

We thank all reviewers for their constructive comments. We appreciate the reviewers for recognizing the novelty and empirical evaluation of our work i.e., "the approach is novel, addresses an important gap in current VLMs' spatial reasoning capabilities" (K1EW), "experiments are comprehensive" (C3yx), and "validating on both spatial VQA and robotic tasks" (C3yx, gF9G).

Following reviewers' suggestions, we provide some additional experiments (results are given in the PDF file) to make the analysis more comprehensive, including:

- **Reviewer gF9G**: Qualitative spatial VQA on ScanQA dataset (similar to qualitative IaOR-VQA) compared to Chat-3D and Chat-3D-v2 (Table. T1).
- **Reviewer K1EW**: Qualitative and quantitative IaOR-VQA on new open source VLM Llama 3.1-70B (Table. T2 and T3).
- **Reviewer K1EW**: Detailed inference speed of our framework in seconds (Table. T4).
- **Reviewers gF9G and C3yx**: New robotics experiment on the task of peg insertion in comparison to SpatialVLM and VoxPoser (Table. T5).

We also provide answers to each concern in the following individual responses. We greatly appreciate the reviewers for their insightful and valuable comments. We believe all the individual questions have been addressed, but we are happy to address any further comments and questions from the reviewers.

---

### Decision · Program_Chairs · 2024-09-25

**Decision:**

Accept (poster)

**Comment:**

The paper presents SpatialPIN, a framework that enhances the spatial reasoning of Visual Language Models (VLMs) by integrating 3D priors from foundation models without additional training. SpatialPIN improves performance in spatial Visual Question Answering (VQA) and robotics tasks like pick-and-stack and trajectory planning. The proposed method and results are reasonable, while there are remaining concerns on the novelty and scalability of the work. After carefully reading the paper, reviews, rebuttals, the AC recommends agreeing with the majority of the reviewers on accepting the paper.